# Neuroticism and Emotional Intelligence in Adolescence: A Mediation Model Moderate by Negative Affect and Self-Esteem

**DOI:** 10.3390/bs12070241

**Published:** 2022-07-19

**Authors:** Nieves Fátima Oropesa Ruiz, Isabel Mercader Rubio, Nieves Gutiérrez Ángel, María Araceli Pérez García

**Affiliations:** Department of Psychology, University of Almería, 04120 Almería, Spain; foropesa@ual.es (N.F.O.R.); aracelip@ual.es (M.A.P.G.)

**Keywords:** adolescence, affect, emotional intelligence, neuroticism, self-esteem

## Abstract

Different studies have revealed that high scores in neuroticism relate to feelings of guilt, flustering, low self-esteem, and insecurity in relationships with others. The main objective of this work is to analyze the relationship between neuroticism and emotional intelligence in the adolescent stage and try to go one step further in the study of that relationship through the formulation of a moderate mediation model in which negative affect participates as a mediating variable and self-esteem as a moderating variable. Method: The total number of adolescents amounted to 742, with a very similar sample in both sexes, 45.1% boys and 51.5% girls. They were between 13 and 19 years old (*M* = 15.63, *SD* = 1.244) and lived in the province of Almería, Spain. Results: First, our main results significantly reflected that the neuroticism personality trait increased negative affect as self-esteem decreased. Second, statistical analyzes showed that the effect of neuroticism on attention and emotional repair was mediated by negative affect, the effect being greater when self-esteem was lower. Therefore, negative affect was presented as a mediating variable in the relationship between neuroticism and attention and emotional repair, with self-esteem acting as a moderating variable in the model. Conclusions: These findings have implications for professional practice with adolescents, since they highlight the importance of carrying out interventions that contribute to the development of self-esteem in the prevention of neuroticism since these actions can help moderate the effect exerted by the negative affect on emotional attention and repair, improving the emotional intelligence of the adolescent and, therefore, their psychological health.

## 1. Introduction

The theoretical model of personality, also known as the *Big Five*, corresponds to a current widely recognized model [1], which is based on a total of five dimensions of one’s own personality: neuroticism (N), extraversion (E), openness (O), agreeableness (A), and conscientiousness (C). In turn, each of these dimensions is made up of a total of six subfactors, which provides us with a detailed analysis of basic tendencies and personality profiles. In this study, we are going to focus on the first of them, neuroticism (N). This construct can refer to frailty, shyness, and emotional instability [2]. It ranges from emotional stability to negative affectivity such as tension, nervousness or anxiety [3], being this the theoretical approach used in this work. In fact, different studies have shown the existence of a positive and direct correlation between perceived stress and anxiety with neuroticism (N) [4,5] (among others).

In the same line as these contributions, we also find works that refer to neuroticism as one of the main contributors to the feeling of psychological distress or depression [6]. In this sense, it is considered that people with high scores in neuroticism have a higher tendency to experience negative emotions, which leads them to have feelings of anxiety, anger, guilt or worry, which also lead to psychological distress and stress [5]. Therefore, neuroticism can be translated into high levels of mean negative affect and variability of negative affect [7] (Wenzel et al., 2022). Some authors such as Muir et al. [8] consider that negative affect is maintained even in advanced adulthood.

Studies related to this model have also revealed that people who score high on the neuroticism subscale have a personality characterized by feelings of guilt, being easily flustered, low self-esteem, and insecurity in relationships with others [9,10,11,12]. They are people who carry out a negative self-assessment of themselves [13] and who present fluctuations in the state of self-esteem—that is, they show instability in self-esteem [14]. In addition, neuroticism is linked to a greater sensitivity to social evaluations, which means that people with a high level of neuroticism react more intensely to situations in which they are negatively judged by others [15] since they are more concerned with how others might see them, which in turn might cause stronger reactions to different perceptions of social inclusion [16].

On the other hand, emotional intelligence is considered a network of skills that allow us to analyze, use, understand and dominate both our own and others’ emotions [17]. In this regard, there are two major conceptualization models of this concept: ability models and mixed models. In this work, we focus on ability models and, specifically, on the model of Mayer and Salovey [18], from which emotional intelligence (EI) is understood as a capacity similar to cognitive intelligence, contemplated as an aptitude or execution [19], leaving aspects related to personality out of this definition. Currently, this model has established itself as one of the most important in terms of the study of emotional intelligence, corresponding to the one with the greatest acceptance and number of investigations in the educational field [20]. Regarding its structure and composition, the model of Mayer and Salovey (1997) [18] is composed of four sections or branches, which allude to the reflexive regulation of emotions, the understanding and analysis of emotions, the facilitation of thought and the perception, assessment and expression of emotion [18,21].

In turn, emotional intelligence has different patterns that are associated with other psychological constructs, such as personality [22], and such facts have been demonstrated by studies that have found high and significant correlations between the Big Five and emotional intelligence [23]. However, regarding this relationship, we must clarify that while Extraversion, Openness, Agreeableness or Conscientiousness result in a positive relationship with emotional intelligence, in the case of neuroticism, the relationship is negative. In that sense, the studies carried out theorize about the strong negative relationship that exists between both psychological constructs, even considering emotional intelligence as the absence of it [24], demonstrating that emotional intelligence is a predictor of well-being, psychological health and negative affect [25].

If we analyze each of the branches of emotional intelligence (attention, Clarity and emotional Regulation), there are numerous studies that have tried to analyze the role of emotional intelligence as a protective factor for different psychological pathologies [26]. However, there are few studies that have aimed to analyze the relationship between each of the dimensions of emotional intelligence and neuroticism [27], moreover, the results found are sometimes somewhat contradictory. Some studies indicate that high levels of emotional Clarity, and not only emotional attention, can hinder recovery in situations of emotional distress [28]. In fact, at a clinical level, the consideration of these aspects is becoming a way of intervening so that the person achieves an improvement in psychological well-being [29]. Other studies find that negative emotions and anxiety or fear correlate with both high and low emotional clarity [30], while other studies show that only low emotional Clarity correlates with neuroticism and negative emotions [31].

Regarding emotional regulation, we must highlight that it has proven to be one of the most important tools both to reduce and control negative emotions [32]. The recent study by Yildirim et al. [27] with a sample of non-clinical adolescents shows that both attention and emotional clarity play a very important role in emotional regulation. Specifically, the results of this work show that, in the relationship between neuroticism and depression, it is the combination of low attention and high emotional clarity that adaptively contributes to emotional self-awareness.

Having said that, the main objective of this work is to analyze the relationship established between neuroticism and emotional intelligence in the stage of adolescence, trying to go one step further in the study of said relationship through the formulation of a moderate mediation model in which negative affect participates as a mediating variable and self-esteem as a moderating variable. This approach aims to explain how it occurs, or in other words, through what psychological mechanisms neuroticism influences emotional intelligence and under what circumstances. Based on this objective, and considering the main findings of the previous scientific literature reviewed, the main research hypotheses are formulated below (Figure 1):
 **H1.***Neuroticism (N) will increase negative affect (NA) as self-esteem decreases.*
 **H2.***The effect of neuroticism (N) on emotional intelligence (EI) (attention, clarity and emotional repair) will be mediated by negative affect (NA), the effect is greater when self-esteem (S-E) is lower.*

## 2. Method

### 2.1. Participants

The total number of adolescents amounted to 742, with a very similar sample in both sexes, 45.1% boys and 51.5% girls. They were between 13 and 19 years old (*M* = 15.63, *SD* = 1.244) and lived in the province of Almería, Spain. Regarding the level of schooling, of the total 68.6% attended compulsory secondary education, with 29.4% enrolled in third grade and 39.2% in fourth grade, and 31.4% were in the process of obtaining a bachelor’s degree, with 14.2% enrolled in first and 17.2 % in second year.

### 2.2. Instruments

-Big Five Inventory (BFI) [3,33]. This questionnaire, made up of 44 items, evaluates the five major personality factors: extraversion (8 items), agreeableness (9 items), conscientiousness (9 items), neuroticism (8 items) and openness to experience (10 items) in a Likert-type scale from 1 (totally disagree) to 5 (totally agree). This research focuses on the neuroticism scale. Neuroticism ranges from emotional stability to negative affectivity such as anxiety, nervousness, sadness or tension (e.g., “He is depressed, melancholic”). The test shows adequate psychometric properties in young people and adults [34,35]. In this study, the reliability values for neuroticism scale is acceptable (α = 0.748).-Trait Meta-Mood Scale (TMMS-24) [36]. It is an adaptation of the original scale Trait Meta-Mood Scale (TMMS) elaborated by Salovey et al. [37]. It measures perceived emotional intelligence through 24 items, expressed on a Likert scale from 1 (totally disagree) to 5 (totally agree). It assesses three dimensions of emotional intelligence: emotional attention (8 items), emotional clarity (8 items) and emotional regulation (8 items). Emotional attention (e.g., “I usually care a lot about how I feel”), refers to the ability to feel and express emotions appropriately. Emotional clarity evaluates the perception that one has about the understanding of one’s own emotional states (e.g., “I can often define my feelings”). Emotion regulation measures the perceived ability to regulate one’s own emotional states correctly (e.g., “Even though I feel bad, I try to think of pleasant things”). Various studies have analyzed the adequacy of its psychometric characteristics with the adolescent [38] and university population [39]. In this work, optimal reliability indices have been obtained in each of the scales: emotional attention (α = 0.896), emotional clarity (α = 0.880) and emotional regulation (α = 0.852).-Rosenberg Self-Esteem Scale [40]. This scale is made up of 10 items, with which a global self-esteem score is obtained, understood as feelings of personal worth and self-respect, through statements such as “I feel that I am a person worthy of affection, at least to the same extent as the others”, to which the adolescents must answer on a Likert-type scale from 1 (totally disagree) to 4 (totally agree). Studies with samples of adolescents have shown the adequate psychometric properties of this scale [41]. In the present work, internal consistency of α is obtained = 0.874.-Positive and Negative Affect Schedule (PANAS) [42]. This test has been used to assess affect. It presents two scales, one for positive affect (pj “Excited”) and another for negative affect (pj, “Distressed”), which are made up of 10 items each, to which adolescents must answer on a Likert-type scale from 1 (very little or not at all) to 5 (extremely). The scores can range between 10 and 50, in order for high scores on each scale to suggest a high presence of positive or negative emotions in the subject, respectively. There is some robust evidence about the proper functioning of the instrument in the adolescent and adult population [43,44]. In this study, the alpha indices have been found to be adequate for both the positive affect scale (α = 0.758) and the negative affect scale (α = 0.759).

### 2.3. Process

Regarding data collection, we contacted the directors of a total of six secondary education centers that participated in the study. They were informed of the objectives, methods, and use of the data in order to provide their informed consent. Once the sessions were scheduled, the research team went to the schools to administer the questionnaires. The tests were applied in the usual classroom assigned to each group in the presence of their teacher. The students were told that their participation was voluntary, they were given the necessary instructions and they were informed of the confidentiality and anonymity in handling the data. All materials and procedures were approved by the Bioethics Committee of the University of Almería.

### 2.4. Statistical Analysis

The data were analyzed using version 27 of the statistical package for the Social Sciences (SPSS) and through version 4.1 of the PROCESS macro [45,46]. First, bivariate descriptive and correlational analyzes were performed between the study variables using Pearson’s *r* Correlation coefficient. Pearson’s correlation coefficient was interpreted as follows: *r* = 0.10 and *r* = 0.23 small effect size, *r* = 0.24 and *r* = 0.36 medium effect size, *r* = 0.37 or more large effect size [47]. Subsequently, the moderate mediation model was calculated for the relationship between neuroticism and emotional intelligence, with self-esteem acting as a moderating variable and affect as a mediating variable, using PROCESS model 7 [48]. Confidence intervals have been calculated on the basis of 10,000 samples.

## 3. Results

### 3.1. Descriptive and Correlational Analysis

Table 1 shows the means, standard deviations and Pearson correlations for the variables studied. Pearson’s correlation analyzes revealed, on the one hand, that the neuroticism personality trait correlated with attention (*r* = 0.304, *p* < 0.01), clarity (*r* = −0.308, *p* < 0.01), and emotional repair (*r* = −0.364, *p* < 0.01), in this case the effect size being large, self-esteem (*r* = −0.343, *p* < 0.01) and Negative Affect (*r* = 0.570, *p* < 0.01), with large effect size. On the other hand, the emotional attention dimension correlated significantly with negative affect (*r* = 0.300, *p* < 0.01), emotional clarity correlated with self-esteem (*r* = 0.318, *p* < 0.01) and negative affect (*r* = −0.201, *p* < 0.01), emotional repair correlated with self-esteem (*r* = 0.338, *p* < 0.01) and negative affect (*r* = −0.308, *p* < 0.01). Finally, self-esteem correlated significantly with negative affect (*r* = −0.397, *p* < 0.01), with a large effect size.

### 3.2. Moderate Mediation Analysis

Starting from the results of the correlational analysis (see Table 1) and taking into account the assumptions of the mediation and moderation analyses, a moderate mediation analysis is carried out on the relationship between the personality trait of neuroticism and emotional intelligence, considering affection as a mediating variable and self-esteem as a moderating variable of said relationship. To carry out the moderated mediation analysis, model 7 of the [rocess macro was used [46,48].

Regarding the emotional attention dimension, as shown in Table 2, the results revealed that, in a first step, neuroticism positively influenced negative affect (β = 4.71, SE = 0.77, *p* < 0.001), in a second step, self-esteem did not influence negative affect (β = 0.08, SE = 0.08, *p* > 0.05) and in a third step, the interaction between neuroticism and self-esteem negatively influenced negative affect (β = −0.07, SE = 0.02, *p* < 0.01).

Conditional effects analysis revealed that neuroticism significantly increased negative affect as self-esteem decreased (Table 3). The moderate mediation index was −0.0286, SE = 0.0152, 95% CI = [−0.0635, −0.0048], revealing that the effect of neuroticism on emotional attention was mediated by Negative Affect, being the effect greater when self-esteem was lower (Table 3). The results of the application of the model can be seen in Figure 2.

In relation to the emotional clarity dimension, the results showed that, in a first step, neuroticism positively influenced negative affect (β = 4.51, SE = 0.79, *p* < 0.001), in a second step, self-esteem did not influence negative affect (β = 0.05, SE = 0.08, *p* > 0.05), in a third step, the interaction between neuroticism and self-esteem negatively influenced negative affect (β = −0.06, SE = 0.02, *p* < 0.01) (Table 3). The moderate mediation index was 0.0160, SE = 0.0119, 95% CI = [−0.0013, 0.0435], indicating that the association between neuroticism and emotional clarity was not moderated by self-esteem (low, medium, or high) in the emotional clarity dimension (Table 4).

Regarding the emotional repair dimension, the results showed that, in a first step, neuroticism positively influenced negative affect (β = 4.66, SE = 0.78, *p* < 0.001). In a second step, self-esteem did not influence in negative affect (β = 0.06, SE = 0.08, *p* > 0.05). In a third step, the interaction between neuroticism and self-esteem negatively influenced negative affect (β = −0.07, SE = 0.02, *p* < 0.01) (see Table 5 and Figure 3). Conditional effects analysis revealed that neuroticism significantly increased negative affect as self-esteem decreased (Table 3). The moderate mediation index was 0.0255, SE = 0.0141, 95% CI = [0.0029, 0.0579], evidencing that the effect of neuroticism on emotional repair was mediated by negative affect, being the effect greater when self-esteem was lower (Table 3). The results of the application of the model can be seen in Figure 3.

## 4. Discussion

In this large-scale study of 742 adolescents, the main objective was to examine the psychological mechanisms by which the relationship between neuroticism and emotional intelligence occurs and under what circumstances. In this regard, two important novelties were provided by this work that should be highlighted. On the one hand, and given the scarcity of previous empirical evidence, this research expanded the study of the relationship between neuroticism and the emotional intelligence dimensions. On the other hand, and regarding the relationship between both constructs, a mediation model moderated by Negative Affect and self-esteem was contributed when no previous empirical studies were found on that matter. In the paragraphs below, the obtained results are discussed, determining whether the hypotheses formulated were confirmed or not relating them to previous research.

In the first place, our main results significantly reflected that neuroticism increases Negative Affect as self-esteem decreases. With these data, our first hypothesis is confirmed, which posited the role of self-esteem as a moderating variable in the relationship between neuroticism and negative affect, in line with previous studies where high levels of neuroticism correlated with low self-esteem [9,10,11,12,49].

Second, the findings of the present investigation revealed that the effect of neuroticism on attention and emotional repair is mediated by negative affect, as self-esteem is lowered. These data have not been significant for the emotional clarity dimension. Other studies, more focused on the relationship between emotional intelligence and emotional distress [28], showed that high levels of emotional attention have been related to greater emotional distress [28]. Regarding the dimension of emotional repair, the results obtained by different researchers are in the same direction as those found in the present investigation [27,32], although these studies do not analyze the role mediator of negative affect in the relationship between neuroticism and emotional intelligence, so our results are revealing at an explanatory and empirical level in this sense. This work probably supports the results of Aluja [49] on the neuroticism construct, allowing us to know how its components operate psychologically (mediating or moderating) in our case, in the explanation of the relationship between neuroticism and emotional intelligence. Therefore, our findings are especially relevant for intervention in the adolescent stage by explaining how these psychological variables are structurally related.

However, as we commented a few lines above, the proposed moderate mediation model does not hold for the emotional clarity dimension. With these data, we partially confirm our second hypothesis. In this regard, the findings of Yildirim et al. [27] in their study on the relationship between neuroticism and depression, showed that low levels of emotional attention, in combination with high levels of emotional clarity, facilitated emotional self-awareness, a finding that should be further explored. Regarding emotional clarity, in our opinion, we hypothesize that perhaps emotional clarity responds to different psychological mechanisms, which may involve other combinations of the variables studied, to the mechanisms of attention and emotional repair, an assumption that requires the realization of new research studies in this regard.

This research step further tries to advance in the study of the mechanisms that intervene in the relationship between personality and emotional intelligence, finding significant results in the particular relationship between neuroticism and emotional attention and repair. To this end, a structural equation model has been validated that incorporates mediation and moderation processes between variables, which has allowed us to analyze the role of negative affect as a mediating psychological variable in the relationship between neuroticism and emotional intelligence, with self-esteem acting as a moderator variable. However, we must be cautious in generalizing the results achieved, since it is a cross-sectional study that should be supported by results from longitudinal studies. In any case, the sample of adolescents participating in the study is broad and the statistical analysis used in the data analysis is robust. Likewise, it should also be considered that the instruments to measure the variables studied are self-report measures, so social desirability biases may be incurred. In this sense, although it is an evaluation technique widely accepted by researchers in the study of adolescence, it could be completed with other psychometric strategies such as observation and interview and also include other significant people in the evaluation process. In addition, to reduce variability, together with standardized questionnaires, physiological measures could be used (evaluation of heart rate, electrodermal activity, among others), which would allow taking into account a person’s biological background, which is considered an important part that leads to different affective experiences. This would be a relevant research field to complete the present study.

## 5. Conclusions

The present work expands the field of previous research on the role of Negative Affect as a determining mechanism in the impact of self-esteem on attention and emotional repair. In this sense, our study makes a significant contribution in this area. In the first place, it proposes that it is possible to reduce negative affect, in situations of neuroticism, through the development of high self-esteem. Second, it proposes a mediation process in the relationship between neuroticism and emotional intelligence, which allows us to conclude that negative affect intervenes as a mediating variable, increasing emotional attention or decreasing emotional regulation, depending on the case, of this relationship. These findings have implications for professional practice with adolescents since they highlight the importance of carrying out educational or therapeutic interventions that contribute to the development of self-esteem in the prevention of neuroticism, since these actions can help moderate the effect that exerts negative affect on emotional attention and repair, improving the emotional intelligence of the adolescent and, therefore, the psychological health.

## Figures and Tables

**Figure 1 behavsci-12-00241-f001:**
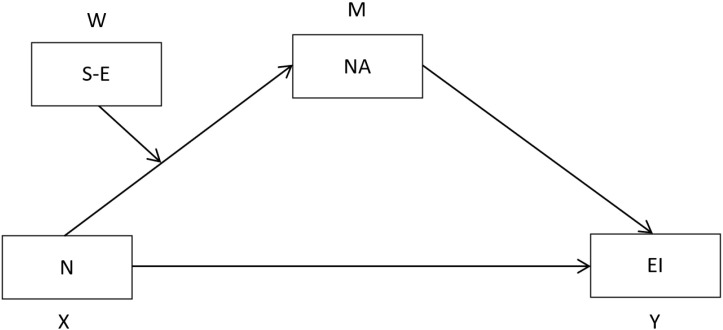
Hypothesized model of moderate mediation.

**Figure 2 behavsci-12-00241-f002:**
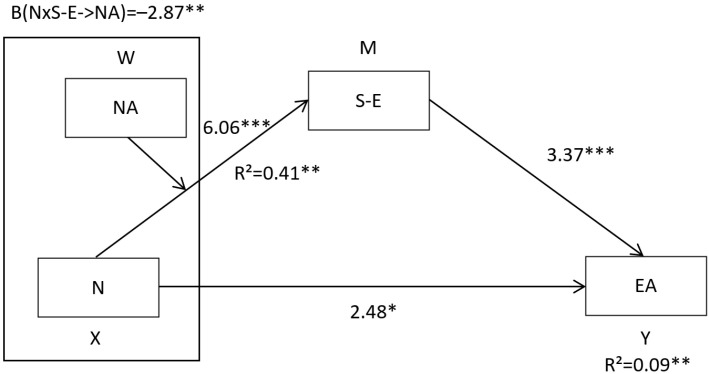
Moderate mediation model for emotional attention. Note. The figure shows the non-standardized regression coefficients (β). * *p* < 0.05; ** *p* < 0.01; *** *p* < 0.001.

**Figure 3 behavsci-12-00241-f003:**
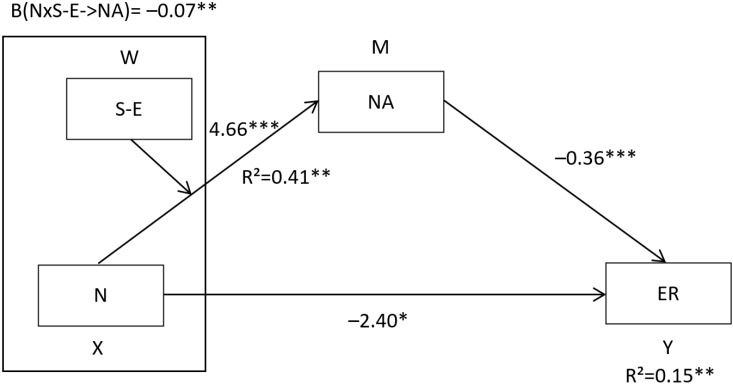
Moderate mediation model for emotional repair. Note. The figure shows the non-standardized regression coefficients (β). * *p* < 0.05; ** *p* < 0.01; *** *p* < 0.001.

**Table 1 behavsci-12-00241-t001:** Bivariate correlations between neuroticism, emotional intelligence, self-esteem and Negative Affect.

	1	2	3	4	5	6
1. neuroticism						
2. emotional attention	0.304 **					
3. emotional Clarity	−0.308 **	0.187 **				
4 emotional repair	−0.364 **	131 **	0.381 **			
5. self-esteem	−0.343 **	−0.057	0.318 **	0.338 **		
6. Negative Affect	0.570 **	0.300 **	−0.201 **	−308 **	−0.397 **	
*M*	2.95	25.69	24.40	25.96	30.49	23.56
*SD*	0.75	7.661	7.32	7.41	6.42	3.65

Note. ** *p* < 0.01.

**Table 2 behavsci-12-00241-t002:** Moderate Mediation Analysis Results for emotional attention.

Variables	Model 1		Model 2	
	β	t	β	t
neuroticism	4.71	6.06 ***	1.45	2.48 *
Self-esteem	0.08	1.01		
neuroticism × self-esteem	−0.07	−2.87 **		
negative affect			0.39	3.37 ***
R^2^	0.41		0.09	
F	95.52 ***		21.15 ***	

Note. * *p* < 0.05; ** *p* < 0.01; *** *p* < 0.001.

**Table 3 behavsci-12-00241-t003:** Conditional and unconditional indirect effects of neuroticism on emotional intelligence, through Negative Affect.

	Effect	Boot SE	IC 95%
Emotional attention			
Self-esteem	Effects conditional		
23.44	3.00	0.25	[2.49, 3.50]
31.00	2.44	0.20	[2.03, 2.85]
37.00	2.00	0.28	[1.04, 2.55]
Self-esteem	Effects unconditional		
23.44	1.17	0.37	[0.47, 1.92]
31.00	0.95	0.29	[0.39, 1.55]
37.00	0.78	0.25	[0.31, 1.32]
Emotional repair			
Self-esteem	Effects conditional		
23.44	3.05	0.26	[2.53, 3.57]
31.00	2.49	0.20	[2.08, 2.89]
37.00	2.06	0.27	[1.52, 2.61]
Self-esteem	Effects unconditional		
23.44	−1.10	0.37	[−1.88, −0.39]
31.00	−0.90	0.30	[−1.51, −0.32]
37.00	−0.75	0.26	[−1.28, −0.26]

**Table 4 behavsci-12-00241-t004:** Results of the moderated mediation analysis for emotional clarity.

Variables	Model 1		Model 2	
	β	t	β	t
Neuroticism	4.51	5.69 ***	−2.13	−3.76 ***
Self-esteem	0.05	0.70		
Neuroticism × self-esteem	−0.06	−2.65 **		
negative affect				−2.07 *
R^2^	0.40		0.09	
F	93.12 ***		20.80 ***	

Note. * *p* < 0.05; ** *p* < 0.01; *** *p* < 0.001.

**Table 5 behavsci-12-00241-t005:** Results of the moderated mediation analysis for emotional repair.

Variables	Model 1		Model 2	
	β	t	β	t
Neuroticism	4.66	5.94 ***	−2.40	−4.41 ***
Self-esteem	0.06	0.85		
Neuroticism × self-esteem	−0.07	−2.74 **		
negative affect			−0.36	−3.38 ***
R^2^	0.41		0.15	
F	97.51 ***		37.47 ***	

Note. ** *p* < 0.01; *** *p* < 0.001.

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
