# Peer review of "Neuroticism and Emotional Intelligence in Adolescence: A Mediation Model Moderate by Negative Affect and Self-Esteem"

_behavsci, 2022, doi:10.3390/bs12070241_

Round 1

Reviewer 1 Report

The paper is well written. Design and content clearly stated. Analysis clear and complete. Discussion appropriate.

To reduce variability, besides standardised questionnaires physiological measures could be used, to find an individual level of emotional reactions, in this respect Heart Rate Variability, Heart Rate, and Electrodermal activity could be used in a standard experimental situation. This aspect could be added at the end of the discussion to point to the fact that the biological background of a person is an important part leading to different affective experiences.

Author Response

Thank you very much for your contributions, which have undoubtedly helped to improve this manuscript. As proposed, we have included at the end of the discussion, in the section on limitations in the measures used, the proposed questions referring to the role of biological factors in affective experiences.

Reviewer 2 Report

The article is well-written scientific work with  psychological and mental health-promotional implications for practical professionals as well. The article meets all scientific requirements, the limitations are well presented. 

My only concern is the overlap between the studied constructs. According to Aluja (2010) low self esteem and negative emotions of depression are subfactors of neuroticism. In this article, self-esteem and negative affects are treated as separate constructs. I suggest clarification of how measurement tools provided measurement of distinct constructs or are there items that are similar in meaning. The article states that neuroticism is measured by negative affectivity such as anxiety, nervousness, sadness or tension. On the other hand, "negative affect scale" of PANAS also measures negative affectivity. 

The two scales correlate with each other at .57 correlational coefficient. I suggest item-level correlations and exclusion of highly correlating items. 

Author Response

Thank you very much for your contributions in improving this work, without a doubt they have made us reflect and clarify the constructs used. The definition of neuroticism is added and improved in the introduction according to the scale used in this study (it does not include low self-esteem, depression, items from the Tellegen questionnaire on negative affect, to define the construct as in the Aluja et al. questionnaire, 2010). Following his suggestions and reflections, Aluja's reference has been included in line 288 to support the relationship between high neuroticism and low self-esteem. Likewise, a brief paragraph (lines 300-304) is added to the discussion in relation to the findings of Aluja et al. (2010) and those obtained in this work.